# Examining the effects of time of day and sleep on generalization

**Marlie C. Tandoc**[1]*, **Mollie Bayda**[2,3], **Craig Poskanzer**[2,4], **Eileen Cho**[2], **Roy Cox**[2,5], **Robert Stickgold**[2], **Anna C. Schapiro**[1,2]*

1 Department of Psychology, University of Pennsylvania, Philadelphia, Pennsylvania, United States of America, 2 Department of Psychiatry, Beth Israel Deaconess Medical Center / Harvard Medical School, Boston, Massachusetts, United States of America, 3 Department of Psychology, University of California-Los Angeles, Los Angeles, California, United States of America, 4 Department of Psychology and Neuroscience, Boston College, Chestnut Hill, Massachusetts, United States of America, 5 Department of Sleep and Cognition, Netherlands Institute for Neuroscience, Amsterdam, The Netherlands

* tandoc@sas.upenn.edu (MCT); aschapir@sas.upenn.edu (ACS)

**Data Availability Statement:** All relevant data are within the manuscript and its Supporting Information files.

**Funding:** This work was supported by: NIH NINDS F32-NS093901 (ACS); MH048832 (RS) The

## Abstract

Extracting shared structure across our experiences allows us to generalize our knowledge to novel contexts. How do different brain states influence this ability to generalize? Using a novel category learning paradigm, we assess the effect of both sleep and time of day on generalization that depends on the flexible integration of recent information. Counter to our expectations, we found no evidence that this form of generalization is better after a night of sleep relative to a day awake. Instead, we observed an effect of time of day, with better generalization in the morning than the evening. This effect also manifested as increased false memory for generalized information. In a nap experiment, we found that generalization did not benefit from having slept recently, suggesting a role for time of day apart from sleep. In follow-up experiments, we were unable to replicate the time of day effect for reasons that may relate to changes in category structure and task engagement. Despite this lack of consistency, we found a morning benefit for generalization when analyzing all the data from experiments with matched protocols (n = 136). We suggest that a state of lowered inhibition in the morning may facilitate spreading activation between otherwise separate memories, promoting this form of generalization.

## Introduction

No two of our experiences are exactly alike. Every decision we make thus involves some degree of generalization. Tracking the shared properties across our experiences allows us to flexibly apply our prior knowledge to novel situations. For example, learning that oil and butter can provide a similar use when frying eggs may allow you to infer that they could support similar functions elsewhere, such as when baking a cake or removing a stuck ring. We will examine how this form of generalization, involving recombinations of previously separate memories, varies as a function of different brain states: offline during sleep, and online at different times of the day.

funders had no role in study design, data collection and analysis, decision to publish, or preparation of the manuscript.

**Competing interests:** The authors have declared that no competing interests exist.

Sleep plays an important role not only in the preservation of our memories, but also in their reorganization [1, 2]. For example, sleep benefits knowledge of shared features [3, 4] and statistical regularities [5] across recently learned items and facilitates the integration of memories into existing knowledge [6, 7]. Sleep may promote the formation of neocortical representations that reflect that shared structure across experiences, which should afford generalization [8].

Despite these findings that sleep can enhance knowledge of structured information, evidence for the benefits of sleep on generalization, the act of applying that knowledge to novel contexts, is more mixed [3, 4, 9–16]. Discovering the kinds of generalization that are enhanced by sleep, and developing behavioral measures that are sensitive to this process, is critical for understanding how sleep-related memory reorganization affects the ability to generalize. We test the idea that generalization across concepts previously seen as unrelated might particularly benefit from the increased representational overlap that emerges during sleep, as they otherwise would be represented distinctly. To this end, we employ a novel category learning paradigm that provides minimal opportunity to extract shared structure during initial learning.

Some forms of generalization, including the integration of information across otherwise separate memories can occur online during a learning experience [17, 18]. The brain states that fluctuate with circadian rhythms, the physiological changes that cycle across the day, may influence this in-the-moment generalization. Within the sleep literature, time of day effects have arisen unexpectedly in tasks that involve generalization. In these studies, categorization, which often involves making decisions about unseen exemplars, was found to be better in the morning [19, 20]. In addition, the tendency to generalize fear extinction responses across contexts is stronger in the morning [15, 21]. The ability to generalize during speech comprehension also appears enhanced in the morning [22]. Together, these studies already provide some evidence that generalization may vary across the day, with superior performance in the morning.

Although these effects were mostly considered secondary curious results, there is a rich literature demonstrating time of day effects on cognition. Several cognitive processes vary across the circadian cycle in both animals [23] and humans [24], including attention [25–27], executive functions [28–39], and various forms of long-term memory [40–53], including semantic memory [54–58].

What leads to time of day effects on cognition? A long-standing framework posits that inhibitory cognitive processes are strongest at peak times of circadian arousal and weakest at off-peak times [32, 34, 39, 59]. These optimal times of day are determined by an individual's chronotype (morning or evening preference) [60], which is typically the evening for young adults [39, 45, 59]). Evidence that off-peak circadian states may be associated with low inhibition include impaired performance on tasks with high cognitive control demands [37, 50, 61] and greater interference between memories [35, 62]. Although interference between memories can be deleterious for their veridical preservation, this lower inhibitory state may be beneficial for connecting memories to uncover shared structure. Indeed, aspects of cognition that rely on noticing connections between otherwise distinct events, such as insight problem-solving in the Remote Associates Test, which involves seeking a word that connects three seemingly unrelated words, show improvements at off-peak times of day [63, 64]. Consistent with theories arguing that reduced cognitive control can be adaptive [65, 66], we suggest that generalization may be another instance where low inhibition is an asset [67]. Specifically, lower inhibition in the morning might facilitate the spreading of activation across different memories, making it easier to see novel connections.

To assess impacts of sleep and time of day on generalization, we developed a variant of a category learning paradigm [4] in which participants learned about three categories of novel satellites. After these satellites were well-learned, participants encountered two *bridge* satellites,

which combined features from two of the initially learned categories. We assessed how encountering examples of satellites that combine features across previously separate categories facilitates the ability to generalize to novel satellites combining features from the same categories. Given that generalization can coincide with false memory formation [68, 69, cf. 70, 71], we also examined false memories for these novel bridge satellites.

Our first three experiments were designed as sleep studies, involving two sessions spaced 12 or 24 hours apart (Experiments 1–3). Although we hypothesized that sleep may benefit this form of generalization, we were unable to detect any evidence of such an effect in these experiments. Instead, we observed a time of day effect, with better generalization in the morning. A nap study suggested that this effect is likely due to the particular time of day, as opposed to having slept recently (Experiment 4). In the next two experiments (Experiments 5 and 6), we pursued this effect using a simplified paradigm and administered additional tasks, the RAT and distractor priming, known to vary with time of day. Finally, we attempted to replicate the time of day effect in our original generalization paradigm, with the RAT and distractor priming task additionally administered at the end (Experiment 7). In these last three experiments, we did not replicate the time of day effect, possibly due to changes to the protocol or to task engagement.

## Methods

### Experiment 1

**Participants.**   33 participants from the Boston area (24 females, age = 21.91, range = 18–34) participated in exchange for monetary compensation. They were randomly assigned to a Morning group or Evening group. For consistency across experiments, conditions are named according to the time of the first session. Data from 3 participants were excluded because in the first test phase they were unable to discriminate novel from trained satellites (mean recognition for novel satellites > mean recognition for trained satellites) or they were unable to reliably recognize trained satellites (mean recognition for trained satellites below baseline), leaving 13 participants in the Morning group and 17 in the Evening group. The same exclusion criteria were implemented in all experiments. Across experiments, differences in sample size between conditions and in some cases limited sample sizes arose due to institution scheduling constraints, such as the availability of required nurse support, and participant cancellations. Participants completed a pre-enrollment screening survey and were only enrolled if they reported: keeping a normal sleep schedule (going to bed before 2am and sleeping at least 6 hours per night); having no psychiatric, neurological, or sleep-related disorders; not taking medications known to interfere with sleep; drinking less than four servings of caffeine per day and less than 10 alcoholic beverages per week; and having normal or corrected-to-normal vision and hearing. The study was approved by the Institutional Review Board of Beth Israel Deaconess Medical Center. All subjects provided written informed consent, and the experiment was carried out in accordance with relevant guidelines and regulations.

**Stimuli.**   The main training stimuli were 15 novel "satellite" objects drawn from three categories, "Alpha," "Beta," and "Gamma" (Fig 1A). Each satellite had five parts: head, back, front leg, back leg, and tail. There were five members of each category, with satellites in the same category sharing most of the five parts. Each category contained one prototype, and each of the other satellites in the category had one part deviating from that prototype. Satellites from different categories did not share any parts. Each satellite had shared features: the "class" name and the parts shared among members of the category; it also had unique features: a unique "code" name and the part unique to that satellite (except for the prototype, which had no

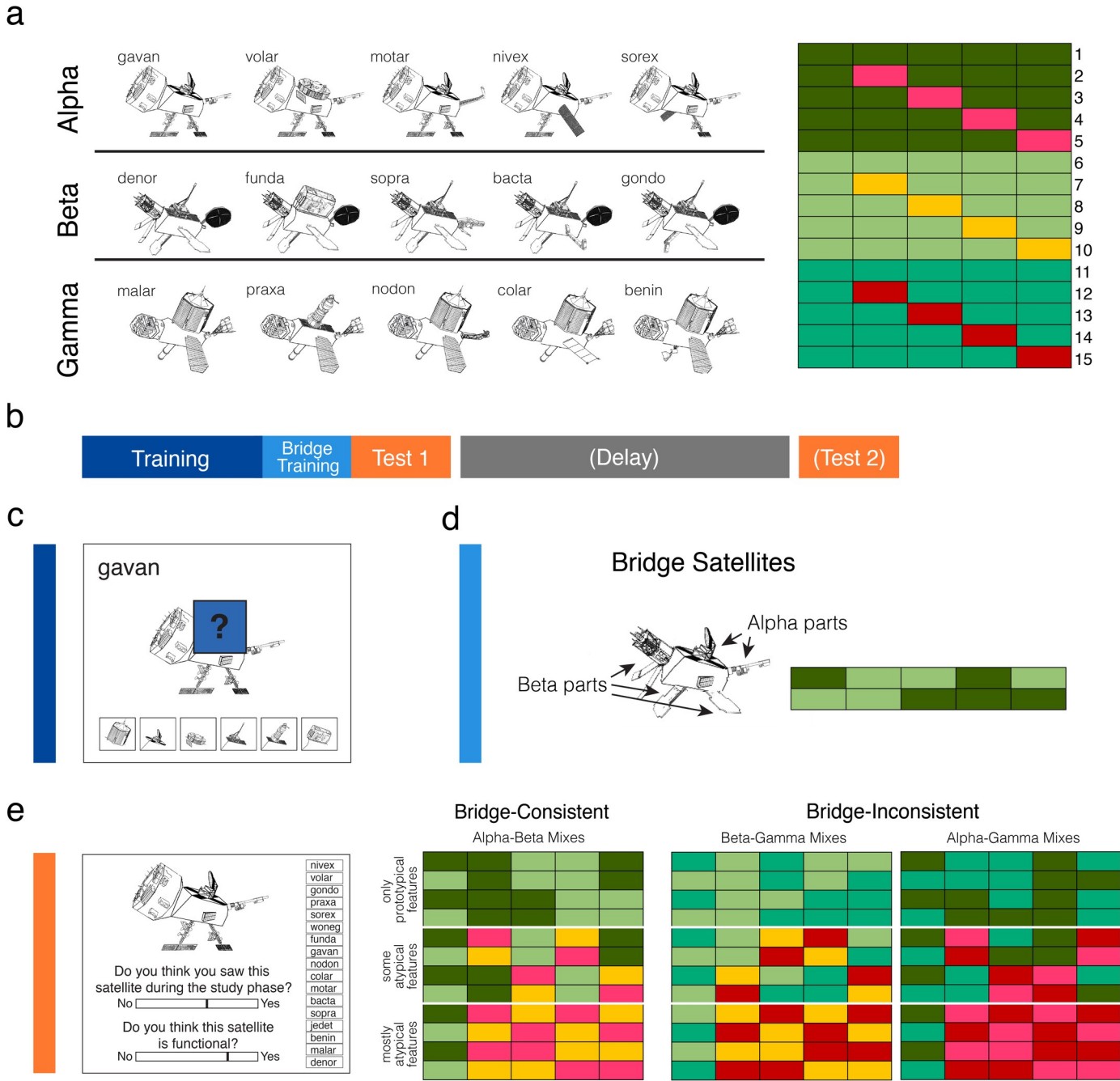

**Fig 1. Stimuli and task design.** (a) Examples of the 15 training exemplars that made up the three categories (left). Category structure of the 15 training exemplars (right) depicting the features (columns) that made up each exemplar (rows). Rows 1–5 are Alpha exemplars, 6–10 Beta, and 11–15 Gamma. Shades of green depict category prototypical features and non-green colors indicate category atypical features. (b) General structure of experiments. (c) Example of a training trial, where participants inferred missing features. (d) Example of a bridge satellite combining prototypical Alpha and Beta features and the feature structure of the two bridge satellites shown during bridge training. Participants learned these satellites in the same way as in the training phase. (e) Schematic of test trial (left) which involves making slider judgments about whether that particular satellite was seen during training (recognition) and if the satellite was functional (functional judgment). Participants might then guess the exemplar code name by clicking on the name (code name memory). Feature structure is depicted (right) for the test items which combined features of the initially learned categories. These satellites could combine features from the same two categories shown during bridge training (bridge-consistent) or from other two pairings of categories (bridge-inconsistent). Test items also could vary in the typicality of their features (only prototypical, some atypical, mostly atypical).

unique part). Satellites were constructed randomly for each participant, constrained by this category structure.

Participants also learned about two "bridge" satellites, which combined prototypical features from two of the categories (Fig 1D). One of the bridge items contained three features from one category and two from the other, and vice versa for the second bridge item.

At test, participants encountered the 17 satellites described above, in addition to 36 novel satellites with the structures shown in Fig 1E. These satellites combined features from each pair of categories with varying use of atypical features: four of the mixes involved only prototypical features from the two categories, four involved some atypical features, and four were mostly atypical features. In total, this produced 12 bridge-consistent and 24 bridge-inconsistent satellites as test items. Test items were carefully designed such that the satellites from each of the three types of mixes had equal amounts of feature overlap with the bridge items.

**Procedure.** Participants were asked not to consume alcohol or recreational drugs in the 24 hours prior to the study, and to abstain from caffeine on the day of the study. They were also asked to maintain a regular sleep schedule in the three days prior to the study, with no bedtimes earlier than 10pm or later than 2am. In the Evening condition, participants came to the lab at 8pm for the training phase. Before beginning, they filled out surveys, including a retrospective three-day sleep diary, the Epworth Sleepiness Scale [72], and the Stanford Sleepiness Scale (SSS; [73]. When finished with the training phase, around 9pm, they completed the immediate test. They slept in the lab with polysomnographic (PSG) recording and filled out the SSS again and were tested again at 9am. The PSG data were not analyzed due to lack of behavioral evidence for sleep effects. In the Morning condition, participants came to the lab at 9am for the training phase. At around 10am they completed the immediate test. They left the lab to participate in normal daily activities except they were instructed not to nap, and they came back to the lab at 9:30pm for a delayed test. The general experimental procedure is depicted in Fig 1B.

During the training phase, participants were first taught about the parts belonging to the 15 satellite objects. Participants were told that "these 15 satellites are functional, which means the parts can work properly together." They were also told that the satellites belong to three classes. Participants were first shown each of the 15 satellites one at a time for five seconds each. The display included the class name, code name, and full satellite image.

The experiment then moved on to the main portion of the training phase, a property inference task. On each trial, participants were shown a satellite with its code name displayed as well as four out of five of its visual features (Fig 1C). A blue box with a question mark occluded one of the visual features and participants were given six options at the bottom as to what the occluded feature might be. These options consisted of all possible versions of that feature across the 15 satellites, including the three prototypical and three atypical variants. Participants used the mouse to click on one of the feature options and received immediate feedback about whether their choice was correct, and the occluding box was removed to reveal the true feature. If the choice was incorrect, the trial would start over with shuffled locations of the feature options, and the participant would continue repeating the trial until they chose the correct feature.

Once the correct feature was chosen, the three possible class names, "Alpha," "Beta," and "Gamma," were displayed, and the participant was prompted to choose the correct name. Participants continued to select class names, with feedback as to whether their response was correct, until clicking the correct name.

The training phase proceeded in blocks, with 45 trials per block. Each block contained three trials for each of the fifteen satellites, two testing each satellite's unique feature and one testing a shared feature. Unique features were queried twice as often as shared, since shared

features are reinforced across members of the category and are thus easier to learn. At the end of each block, participants were shown their accuracy for the visual feature guessing and for the class name guessing. They were told that they needed to reach 66% correct across trials on the visual parts and 90% correct on the class names. Once they reached both criteria or one hour had passed, they moved on to the bridge training phase.

In the bridge training phase, participants learned about the two bridge items, which combined features from two of the categories (Fig 1D). Participants were told: "You have now studied 15 satellites. In this next part of the study phase, we are interested in how difficult it will be to now learn about two new satellites. Each of these new satellites has combinations of parts that are functional." This was simply a cover story, as we did not want participants to be clued in to the structure of the bridge items. Participants were then shown the code names and images of the two new satellites (no class names). They then did the same property inference task as before except that they did not guess class names. There were blocks of ten trials to query each of the five features from the two satellites, with trials alternating between the two satellites. They repeated the block until identifying nine out of ten features correctly.

Participants then completed the test phase, which had 53 trials. These trials consisted of the 15 trained satellites and 2 trained bridge items, as well as 36 novel satellites. On each trial, participants saw a satellite with one slider bar below asking if they had seen that satellite during the study phase (Fig 1E). The ends of the scales were labeled "Definitely not" and "Definitely yes" and the middle was labeled "Not sure." They clicked a location on the slider and then either a "submit" button to register the response or a "reset" button, allowing them to change their response before submitting. As long as the slider response was not placed within the bottom 5% of the scale (indicating that they definitely did not recognize the satellite), 17 options for code names appeared on the right side of the screen, and participants clicked to choose an option. They could then submit or reset. Subjects next saw a second slider bar appear below the first, prompting a response on whether the satellite was functional. The following note was always displayed next to this slider: "Please try to make use of the entire slider. (All satellites that you have studied are functional. Many of the satellites that you have not studied are also functional.)" Slider values were recorded in the range -1 (Definitely not) to 1 (Definitely yes).

For the delayed test approximately 12 hours later, participants completed the same test phase with the trials in a different random order. The experiments were implemented in MATLAB with Psychophysics Toolbox [74].

**Generalization index calculation.** For each participant, a generalization index was calculated as a difference score (mean slider ratings for novel bridge-consistent Satellites–mean slider ratings for novel bridge-inconsistent satellites) separately for their functional slider ratings and recognition slider ratings. For functional judgment generalization, a positive index indicates that a participant endorses novel bridge-consistent satellites (satellites that share features from the two categories shown to produce a functional satellite during the bridge phase) as more functional relative to novel bridge-inconsistent satellites (satellites that combine features from categories never shown to produce a functional satellite). For recognition generalization, a positive index indicates that a participant false alarms to novel bridge-consistent satellites more than to novel bridge-inconsistent satellites.

## Experiment 2

31 participants were recruited (18 females, age = 22.23, range = 18–35) and 4 were excluded according to the criteria specified above. This experiment was the same as Experiment 1 except that the number of blocks of exposure for the bridge items was not determined by performance but instead fixed separately for each group in an attempt to match performance between

groups. The Morning group (n = 14) only saw one block of bridge items and the Evening group (n = 13) saw six blocks. Given the effect size for functional judgment generalization observed in Experiment 1 (Cohen's $d$ = 1.20), this sample provides over 90% power to detect the same effect. In Experiment 1 the Morning group took 2.46±0.27 blocks to learn and the Evening group took 2.59±0.32 blocks. Relative to Experiment 1, the Experiment 2 Morning group thus received less bridge training and the Experiment 2 Evening group received more bridge training.

## Experiment 3

This experiment was the same as Experiment 1 except that the delayed test took place 24 hours later instead of approximately 12 hours later. Participants did not sleep in the lab. From this experiment on, we also collected the Morningness-Eveningness Questionnaire [60] at the end of the protocol for each participant (MEQ). 29 participants were recruited (15 females, age = 23.03, range = 18–32). Two participants failed to meet the inclusion criteria, leaving 14 participants in the Morning group and 13 participants in the Evening group.

## Experiment 4

This experiment was the same as Experiment 1 except that all participants began training at 4pm, and there was no delayed test. Participants were randomly assigned to either take a nap in the lab with PSG recording prior to the test (Nap group) or to come into the lab without having taken a nap (No Nap group). 41 participants were recruited (23 females, age = 23.39, range 18–33) and 4 were excluded, leaving 20 participants in the Nap group and 17 in the No Nap group. One participant's total sleep time was manually recorded due to a technical error.

## Experiment 5

This experiment was the same as Experiment 1 except that there was again no delayed test, there were two additional new tasks administered before the satellite task, and the satellite task was simplified to allow time for the additional tasks. Specifically, there were now three satellites instead of five in each of the three categories. Using the numbering shown in Fig 1A, items 3, 5, 7, 9, 12, and 15 were removed. There were now 27 trials per block during the initial training phase. The test was also shortened: All generalization test trials with atypical features were removed, resulting in a total of 21 trials. 58 participants were recruited (43 females, age = 21.68, range = 18–30) and 15 were excluded (1 due to procedural difficulties), leaving 25 in the Morning group and 18 in the Evening group.

The two new tasks were the Remote Associates Task (RAT; [75]) and a distractor priming task [37, 76]. Participants first completed the RAT, in which on every trial they saw three words and had to find a fourth that connects them by forming a two-word phrase or a compound word. Participants were given the example *Eight / Skate / Stick*, with the solution *Figure*. They completed two practice trials followed by 20 test trials. They were given a maximum of 30 seconds to type in a solution word on each trial. If they typed the correct word with errors, partial credit was assigned using Levenshtein distance, a string comparison metric that calculates the minimum number of character changes needed to transition from one word (the typed response) to another (the solution word). Only 2.5% of participant responses were assigned partial credit using this method. Manually scoring these partial credits as either correct or incorrect for semantic accuracy (ignoring spelling and typographical errors) had no material impact on the results.

The distractor priming task had two parts. The first part was a 1-back task, in which participants were presented with a series of red line drawings of familiar objects and animals.

Superimposed on the pictures were words and nonwords in black font. Each stimulus pair was displayed for 1000 ms, followed by a blank screen for 500 ms. Participants were told to ignore the words and to press one key whenever the current drawing was identical to the previous one, and a different key otherwise. After a practice phase, there were 69 trials of the task, with 15 target words, 15 fillers, and 39 nonwords. The identity of the target words and fillers were counterbalanced across subjects. About one in every seven trials had a repeating image.

The second part of the distractor priming task was presented to participants as a completely unrelated task. In this part, participants were told they would see a "mystery" word with some blank letters and to quickly type in the first word that comes to mind to complete the missing letters: "For example, if you see T_BL_, you might type TABLE and then press enter." Participants were given a maximum of 5 s to respond on each trial. There were 15 fragments that could be completed by a target word from the prior task, 15 fragments of words from the other counterbalanced group's targets, and 15 easily-solvable filler fragments. Our implementation of the distractor priming task differed from prior implementations in that there was no delay between the two parts of the task, and participants typed their fragment completion responses instead of speaking them outloud.

To generate a priming score, we compared the typed response to the target word on each trial. If the participant typed the correct target word, the response received a score of 1. If they typed a different word or did not type a word, the response received a score of 0. Otherwise, we used Levenshtein distance to provide partial credit. Priming was then calculated as a difference score for each participant between their mean score to targets and mean score to novel items (which served as targets for the counterbalanced group). As prior studies have calculated this measure slightly differently, contrasting mean scores to targets with means scores on these same words from the counterbalanced group, we also computed the priming score in this manner and found that it did not change the results.

## Experiment 6

This experiment was the same as Experiment 5 except that the RAT and priming test were administered after (rather than before) the satellite task. 53 participants were recruited (42 females, age = 21.69, range = 18–34) and 4 were excluded, leaving 26 participants in the Morning group and 23 in the Evening group. One participant did not complete surveys, but was still included in analyses.

## Experiment 7

This experiment was the same as Experiment 6 except that the original stimuli and test set, with 5 satellites per category and 53 test items, were used for the satellite task. 86 participants were recruited (60 females, age = 21.74, range = 18–36) and 7 were excluded, leaving 41 participants in the Morning group and 38 participants in the Evening group. One participant did not complete surveys and another participant was excluded from the priming test due to procedural difficulties.

Design differences for each experiment, as well as the observed time of day effects on generalization, are summarized in Table 1.

# Results

## Experiment 1

**Training performance and surveys.** Training phase results, including the mean proportion correct on the last block of initial training for feature and class accuracy, the number of

**Table 1. Experiment design and time of day effects on generalization.**

| | Experiment Design | | | | | | | Time of day effect on generalization | |
|---|---|---|---|---|---|---|---|---|---|
| | Number of Sessions | Delay Between Sessions | Session 1 Start Time | Session 2 Start Time | Bridge Exposure | Exemplars Per Category | Number of Test Trials | Order of Tasks | Functional judgment $p$ | Recognition $p$ |
| **Exp. 1** | 2 | 12 hours | Morning: 9AM Evening: 8PM | Morning: 9:30PM Evening: 9AM | To criterion | 5 | 53 | | .003** | .06§ |
| **Exp. 2** | 2 | 12 hours | Morning: 9AM Evening: 8PM | Morning: 9:30PM Evening: 9AM | Morning: 1 block Evening: 6 blocks | 5 | 53 | | .44 | .52 |
| **Exp. 3** | 2 | 24 hours | Morning: 9AM Evening: 8PM | Morning: 9AM Evening: 8PM | To criterion | 5 | 53 | | .0003*** | .01* |
| **Exp. 4** | 1 | | 4PM | | To criterion | 5 | 53 | | | |
| **Exp. 5** | 1 | | Morning: 9AM Evening: 8PM | | To criterion | 3 | 21 | RAT, Distractor Priming, Generalization | .08§ | .18 |
| **Exp. 6** | 1 | | Morning: 9AM Evening: 8PM | | To criterion | 3 | 21 | Generalization, RAT, Distractor Priming | .47 | .34 |
| **Exp. 7** | 1 | | Morning: 9AM Evening: 8PM | | To criterion | 5 | 53 | Generalization, RAT, Distractor Priming | .56 | .93 |
| **Exp. 1, 3, & 7 Combined** | | | Morning: 9AM Evening: 8PM | | To criterion | 5 | 53 | | .0007*** | .04* |

Number of Sessions = number of experimental sessions; two session experiments had a second test administered after a delay. Delay Between Sessions = approximate time between Session 1 and Session 2. Session 1 Start Time = approximate start time of Session 1 for each group. Session 2 Start Time = approximate start time of Session 2 (referred to in text as Test 2) for each group. Bridge Exposure = number of blocks (10 trials per block) that participants encountered bridge satellites during the training phase. Criterion was reached when a participant achieved nine out of ten trials in a block correct. In Experiment 2 the number of blocks was different between groups, reflecting our aim to match initial generalization performance in this experiment. Exemplars Per Category = the number of unique satellites in each of the three categories encountered during the training phase. Number of Test Trials = total number of trials during the test phase in one session. Order of Tasks = the order of the tasks that participants completed in experiments that administered additional measures (RAT, Distractor Priming). Generalization refers to the full generalization paradigm (i.e. both training and test). Functional judgment $p$ = Difference between morning and evening group for functional judgment generalization. Recognition $p$ = Difference between morning and evening group for recognition generalization. ***$p < .001$, **$p < .01$, *$p < .05$, §$p < .1$

blocks required to learn the initial group of satellites as well as the bridge satellites are reported in S1 Table. We did not find any differences between groups. Refer to S2 Table for survey measures for all experiments, including Stanford Sleepiness Scale reports.

**Basic measures of performance in Test 1.** We first assessed performance on functional and recognition judgments for trained satellites (the 17 satellites from the initial and bridge phases). While there was no difference between groups on recognition judgments for trained satellites ($t[28] = 0.36$, $p = .72$), the Morning group rated trained satellites as more functional than the Evening group ($t[28] = 2.79$, $p = .009$; Morning: 0.72±0.05; Evening: 0.55±0.04). The groups did not differ in their mean overall functional ($t[28] = 0.74$, $p = .47$) or recognition ($t$

[28] = -1.58, *p* = .13) slider responses. We found no difference between groups in the proportion of correct responses for code name memory (*t*[28] = 0.07, *p* = 0.94). Overall, basic measures of performance in Test 1 did not differ between groups, except that the Morning group rated trained satellites as more functional.

**Time of day effects on generalization in Test 1.** Before assessing the role of sleep on generalization, we sought to rule out any differences between groups on initial generalization performance in Test 1. Strikingly, functional judgment generalization was significantly different between the groups tested in the morning and evening (*t*[28] = 3.28, *p* = .003) whereby the Morning group was better able to attribute the functional property to novel bridge-consistent satellites than the Evening group (Fig 2A). The Morning group also exhibited marginally higher recognition generalization than the Evening group (*t*[28] = 1.92, *p* = .06), suggesting that participants tested in the morning had more false memories for novel bridge-consistent satellites than those tested in the evening. Furthermore, only the Morning group exhibited positive generalization for functional judgments (Morning: *t*[12] = 3.47, *p* = .005, Evening: *t*[16] = -0.88, *p* = 0.39), indicating that only participants tested in the morning reliably rated bridge-consistent satellites as more functional than bridge-inconsistent ones. Neither group showed reliably positive generalization for the recognition judgment (Morning: *t*[12] = 1.55, *p* = .15; Evening: *t*[16] = -1.09, *p* = .29). There was a positive correlation between functional judgment generalization and recognition generalization (*r* = 0.73, *p* = .000005), suggesting that if a participant was likely to endorse novel bridge-consistent satellites as functional, they were also more likely to falsely endorse novel bridge-consistent satellites as having been seen during

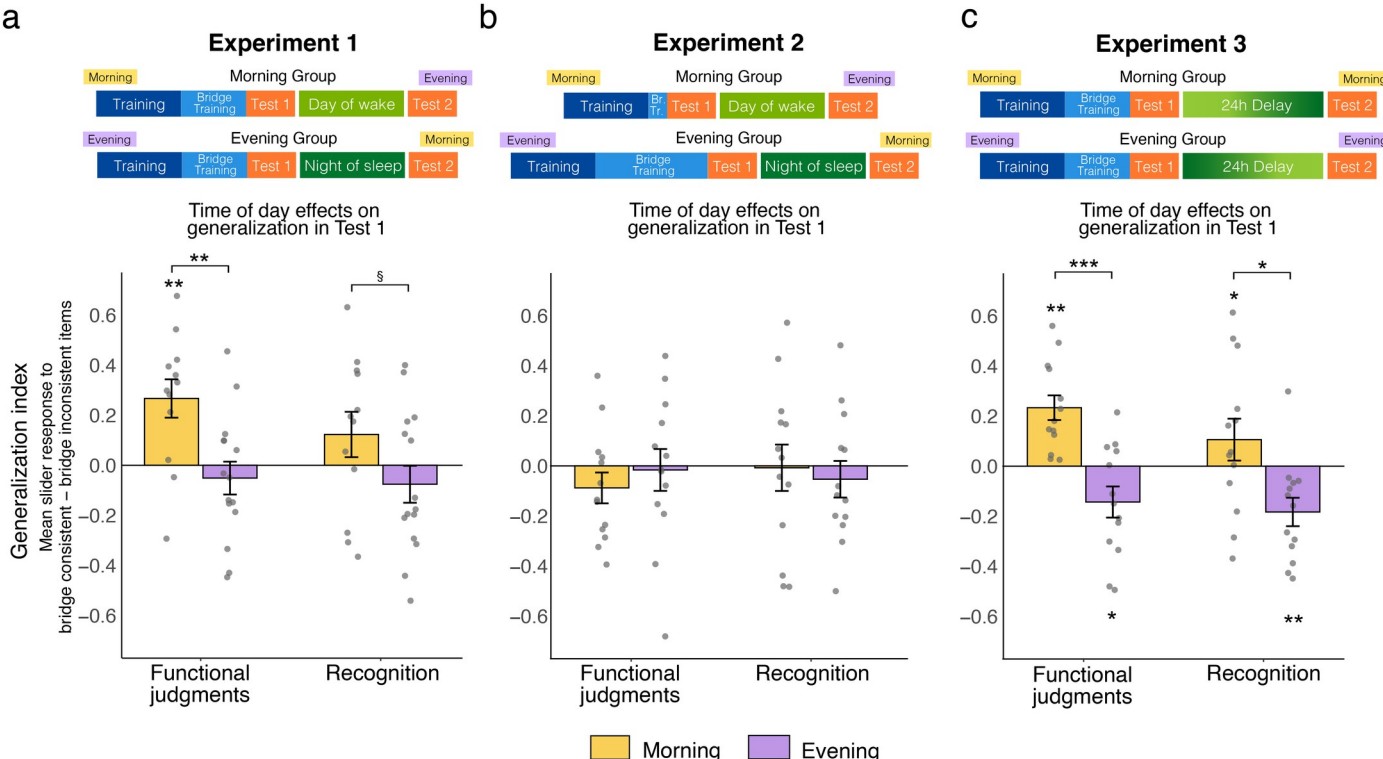

**Fig 2. Time of day effects on generalization in Test 1.** (a) Experiment 1. (b) Experiment 2, where bridge training was fixed to match initial generalization between groups: the morning group received only one block of bridge-satellite training whereas the evening group received six blocks. This is in contrast to Experiment 1 and 3 where participants trained to criterion. (c) Experiment 3. A schematic of the timeline (not to scale) for each experiment is shown above. Gray dots correspond to individual participants and error bars denote ± one SEM. ***p* < .001, ***p* < .01, **p* < .05, §*p* < .1.

training. Time since awakening from sleep did not correlate with either measure of generalization for the Morning or Evening group (*p's* > .57).

The novel bridge satellite test items varied in the typicality of their features (referred to as feature type). Regressions with time of day and feature type (ordered as prototypical, some atypical, mostly atypical) as predictors revealed a significant main effect of time of day on functional judgment generalization ($B = 0.15$, t = 3.51, *p* = .0007) and on recognition memory generalization ($B = 0.10$, $t = 2.15$, *p* = .03). There was no main effect of feature type, nor was there any interaction between time of day and feature type on either measure of generalization (*p's* > .42). These results indicate that the morning-time benefit for generalization was consistent across all bridge-satellite feature types.

**Generalization in Test 2.**   In Session 2, there was no difference in generalization (functional judgments: $t[28] = -0.52$, *p* = .61; recognition: $t[28] = 0.20$, *p* = 0.84) between Morning and Evening groups, and neither group significantly differed from chance (*p's* > .09; S1 Fig). There was a difference between groups in the change from Test 1 to Test 2 in functional judgment generalization ($t[28] = -2.44$, *p* = .02) and a marginal difference for recognition generalization ($t[28] = -1.93$, *p* = .06; S2 Fig). However, these differences cannot be interpreted as sleep effects because of the strong initial difference between groups (the time of day effect described above). Indeed, a multiple regression including initial performance and group as predictors revealed that initial performance negatively predicted the change in generalization performance across the delay ($B = -0.48$, $t = -2.52$, *p* = .02) whereas group was not a significant predictor ($B = 0.06$, $t = 0.10$, *p* = .33), nor was there an interaction ($B = -0.17$, $t = -0.89$, *p* = .38). There was no difference between groups in the change from Test 1 to Test 2 in code name memory ($t[28] = -1.14$, *p* = .26).

**Discussion.**   Although Morning and Evening groups differed in their change in generalization performance across the delay in a manner consistent with a benefit of sleep on generalization, this effect was most likely due to a strong time of day effect observed in the first session, whereby generalization was better in the morning than evening. Other measures of performance such as initial learning and exemplar memory did not differ between Morning and Evening groups, suggesting an effect specific to generalization (although the Morning group also tended to rate trained satellites as more functional).

## Experiment 2

Due to a time of day effect on initial generalization performance in Experiment 1, we were unable to cleanly assess the influence of sleep on generalization. To address this in Experiment 2, we attempted to match initial generalization performance by providing six blocks of bridge training to the Evening group and only one block to the Morning group. This experiment is thus unlike all other experiments, where participants trained to a criterion to ensure that they successfully learned the bridge satellites. Otherwise, the protocol was identical to Experiment 1.

**Training performance.**   The manipulation was successful in changing bridge performance: There was a significant difference between groups on feature accuracy in the last block of bridge training ($t[25] = -4.87$, *p* = .0005; Morning: 0.69±0.06; Evening: 0.98±0.01). Providing the Evening group with six times as much training as the Morning group resulted in superior learning of the bridge satellite features. Other training phase results are presented in S1 Table.

**Basic measures of performance in Test 1.**   Mean slider ratings did not differ between Morning and Evening groups for trained satellites (functional: $t[25] = -0.55$, *p* = .59, recognition: $t[25] = -0.62$, *p* = .54) or for satellites overall (functional: $t[25] = 0.88$, *p* = 0.39; recognition: ($t[25] = 0.92$, *p* = .37). There was also no difference between groups in code name memory ($t[25] = -1.45$, *p* = .16).

**Initial generalization performance in Test 1.**   There was no difference between Morning and Evening groups on either measure of generalization (functional: $t[25]$ = -0.79, $p$ = .44; recognition: $t[25]$ = 0.65, $p$ = 0.52; Fig 2B). Neither group differed from zero on either generalization measure ($p's > .12$). Manipulating the amount of bridge exposure was thus successful in matching initial generalization performance between groups.

**Change in generalization across sessions.**   With matched initial generalization performance, we next examined the change in performance across the delay. Counter to our prediction that 12 hours including sleep (Evening group) should facilitate generalization relative to 12 hours awake (Morning group), we found no difference between groups on the change in generalization performance between Test 1 and Test 2 on either measure of generalization (functional: $t[25]$ = 1.61, $p$ = .12; recognition: $t[25]$ = 0.30, $p$ = .77; S2 Fig). In Session 2 there was again no difference in generalization between groups (functional: $t[25]$ = -0.50, $p$ = .62; recognition: $t[25]$ = -0.81, $p$ = .42; S1 Fig). Neither group was above chance on either generalization measure ($p's > .33$). There was no effect of sleep on code name memory ($t[25]$ = -1.44, $p$ = .16).

**Discussion.**   We successfully matched initial generalization performance by providing the Evening group with much more bridge satellite training than the Morning group (who received very limited bridge-satellite training). This is unlike Experiment 1, where participants learned the bridge-satellites to a performance criterion. Counter to our prediction that a night of sleep should facilitate generalization relative to a day of wake, we found no evidence that sleep had any effect on generalization. Instead, the time of day effect observed in Experiment 1 is echoed in this experiment: even with six times as much bridge training, evening participants are unable to generalize above chance and no differently than the morning participants, who received much less training.

## Experiment 3

In this experiment, our primary aim was to replicate the time of day effect observed in Experiment 1. The only design change was that the delay period between Session 1 and Session 2 was 24 hours instead of 12. We tested after 24 hours to verify that a sleep effect would not emerge when time of day was matched across test sessions. From this experiment onwards, we also assessed chronotype using the Morningness-Eveningness Questionnaire (MEQ; [60] to examine potential interactions between circadian preference and time of day effects. Basic descriptive statistics for each group for the MEQ are provided in S2 Table.

**Basic measures of performance in Test 1.**   There was no difference between groups on mean slider ratings for trained satellites (functional: $t[25]$ = 1.22, p = .23; recognition: $t[25]$ = 0.55, $p$ = .58). However, the Evening group endorsed satellites overall as less functional ($t[25]$ = 4.20, $p$ = .0003; Morning: 0.31±0.04; Evening: 0.06±0.04) and gave lower recognition ratings ($t[25]$ = 2.11, $p$ = .04; Morning: 0.04±0.05; Evening: -0.11±0.05). Evening participants also exhibited marginally better code name memory for trained satellites than Morning participants ($t[25]$ = -1.99, $p$ = .06; Morning: 0.45±0.04; Evening: 0.57±0.05).

**Time of day effects on generalization in Test 1.**   The main goal of this experiment was to test whether the morning-time benefit for generalization would replicate. Indeed, we found a significant difference between Morning and Evening groups on functional judgment generalization ($t[25]$ = 4.15, $p$ = .0003; Fig 2C). The Morning group exhibited reliably positive generalization for functional judgments ($t[13]$ = 3.63, $p$ = .003) indicating that they rated bridge-consistent satellites as more functional than bridge-inconsistent ones. The morning-time benefit for generalization also extended to recognition generalization, with the Morning group showing increased false alarms to novel bridge-consistent satellites than the Evening group ($t$

[25] = 2.65, *p* = .01). The Morning group did not differ from zero on recognition generalization (*t*[13] = 1.02, *p* = 0.33). Notably, the Evening group demonstrated reliably negative generalization on both functional judgments (t[12] = -2.30, *p* = .04) and recognition (*t*[12] = -3.22, *p* = .007), indicating they were more likely to endorse novel bridge-inconsistent satellites as functional and having been seen before relative to novel bridge-consistent satellites. Time since awakening from sleep did not relate to either generalization measure in either group (*p's* > .24).

Regressions with time of day and bridge satellite feature type (prototypical, some atypical, mostly atypical) revealed a main effect of time of day on both functional judgment generalization (*B* = 0.17, *t* = 3.96, *p* = .0002) and recognition generalization (*B* = 0.13, *t* = 2.92, *p* = .005). There was also a main effect of feature type with better functional judgment generalization for satellites that were made up of increasingly more atypical features (*B* = 0.11, *t* = 2.14, *p* = .04). A similar marginal main effect of feature type was observed for recognition generalization (*B* = .11, *t* = 1.93, *p* = .06). Interaction terms did not reach significance (*p's* > .50).

To examine whether circadian preference modulated the observed time of day effects, we ran regression models where we let MEQ score and time of day interact as predictors for generalization. This yielded a significant main effect of time of day on the functional judgment measure of generalization (*B* = 0.19, *t* = 5.15, *p* = .00003), but no interaction between MEQ and time of day (*B* = -0.004, *t* = -1.29, *p* = .21). This suggests that there is an overall benefit for generalization in the morning that is not influenced by an individual's circadian preference. There was also a main effect of MEQ on generalization (*B* = 0.01, *t* = 3.01, *p* = .006), with more strongly morning-type participants exhibiting better generalization. In the model for recognition generalization, there was a morning-time benefit for generalization (*B* = 0.14, *t* = 2.73, *p* = .01), but no main effect of MEQ (*B* = 0.01, *t* = 1.05, *p* = .31) nor an interaction (*B* = -0.01, *t* = -0.54, *p* = .59) Although these findings are consistent with the possibility that circadian preference does not interact with the time of day effects on generalization, it is possible that we did not have sufficient sampling of extreme chronotypes for an influence of circadian preference to emerge in our sample (most of our sample were neutral chronotypes; S2 Table).

**Generalization in Test 2.** We next assessed differences across the 24h delay for the two groups. In Test 2, there was no difference between groups on either measure of generalization, and neither group was above chance (*p's* > .62; S1 Fig). As groups did not differ from zero on the generalization index, the lack of a time of day effect in Test 2 may be due to overall forgetting across the 24h delay. Similar to the results in Experiment 1, there were reliable differences between groups in the change in generalization from Test 1 to Test 2 (functional: *t*[25] = -2.79, *p* = .01, recognition: *t*[25] = -2.14, *p* = .04; S2 Fig). However, only initial performance (*B* = -0.70, *t* = -3.11, *p* = .005), and not group (*B* = 0.03, *t* = 0.43, *p* = .67), predicted the change in generalization across the delay (no interaction: *B* = -0.30, *t* = -1.35, *p* = .19). There was no difference between groups in the change in code name memory between sessions (*t*[25] = 1.01, *p* = .32).

**Discussion.** Experiment 3 replicated the morning-time benefit for generalization observed in Experiment 1. Not only were morning participants better at generalizing the functional property to novel bridge-consistent satellites, but they were also more likely to have false memories of having seen these satellites before.

The Evening group exhibited reliably negative generalization in this experiment, meaning they tended to rate novel bridge-*inconsistent* satellites as more functional and more familiar than novel bridge-consistent satellites. This result may be consistent with non-monotonic accounts of memory plasticity [77, 78]: Moderate instead of strong activation of related satellites during the study of the bridge items, possibly due to higher inhibition in the evening than morning, could result in repulsion instead of integration of the bridged categories.

## Experiment 4

One benefit of sleep is its restorative effect, whereby participants who have slept recently exhibit enhanced memory encoding [79, 80]. Could the morning-time benefit for generalization be driven by the Morning group having slept more recently than the Evening group? In the above experiments, the lack of a relationship between time since awakening and performance in the Morning group provide some evidence contrary to this account. To more directly disentangle a restorative sleep account from a time of day account, we ran a nap study in the afternoon where half the participants had a 90-minute nap opportunity prior to the 4:00PM experiment (Nap group) and the other half did not nap (No Nap group). There was no delayed test in this experiment (or any of the following experiments).

**Basic measures of performance and nap duration.** The mean total sleep time for the Nap group was 72.75 minutes (SE = ±4.09). For the trained satellites, the Nap group provided marginally higher functional ratings ($t$[35] = 1.92, $p$ = .06; Nap: 0.74±0.05; No Nap: 0.62±0.05) and reliably higher recognition ratings ($t$[35] = 2.41, $p$ = .02; Nap: 0.59±0.05; No Nap: 0.42 ±0.05). Across all satellites, there was no difference between groups in mean slider judgments for recognition ($t$[35] = 1.22, $p$ = .23), although the Nap group rated them as marginally more functional ($t$[35] = 1.88, $p$ = .07; Nap: 0.32±0.04; No Nap: 0.18±0.06). There was no difference in code name memory between groups ($t$[35] = 1.00, $p$ = .32).

**Recent sleep effects on generalization.** Generalization did not differ between the Nap and No Nap groups (functional: $t$[35] = -0.83, $p$ = .41; recognition: $t$[35] = -1.47, $p$ = .15; Fig 3). The Nap group did not differ from zero on either measure of generalization (functional: $t$[19] = 0.61, $p$ = 0.55, recognition: $t$[19] = -0.26, $p$ = 0.80). The No Nap group exhibited marginally positive generalization (functional: $t$[16] = 2.05, $p$ = .06, recognition: $t$[16] = 2.06, $p$ = .06). A regression model with group and bridge satellite feature type (prototypical, some atypical, mostly atypical) predicting functional judgment generalization yielded no significant terms ($p$'s > .11). For recognition memory generalization, the model revealed a marginal main effect of group ($B$ = -0.07, $t$ = -1.71, $p$ = .09), with slightly better generalization in the No Nap group, and a marginal main effect of bridge satellite feature type ($B$ = 0.95, $t$ = 1.92, $p$ = .06), with better generalization observed in satellites incorporating more atypical features (similar to Experiment 3). No other model terms were significant ($p$'s > .34). Overall, the results suggest that having napped recently does not benefit generalization.

**Discussion.** Generalization performance did not differ between participants who had a 90-minute nap opportunity prior to the task relative to participants who did not nap. Therefore, the restorative effects of sleep following a nap [79, 80] do not appear to facilitate generalization in our paradigm. This finding is consistent with the lack of relationship in our earlier experiments between time since awakening in the morning and generalization. Thus, the morning-time benefit for generalization cannot be attributed to the Morning groups having slept recently, implicating a more general role of time of day. Counter to work demonstrating enhanced episodic learning following a nap [79, 80], initial learning measures and memory for exemplar names were also unaffected by having slept recently.

## Experiment 5

Experiments thus far demonstrated that generalization is better in the morning than evening (Experiments 1 and 3); that even with six times the amount of bridge satellite training, participants tested in the evening are no better at generalizing than participants tested in the morning (Experiment 2); and that this effect is not likely due to having slept recently (Experiment 4). Taken together, these experiments suggest that the time of day influences this form of generalization. How might generalization relate to other tasks known to be better at non-optimal

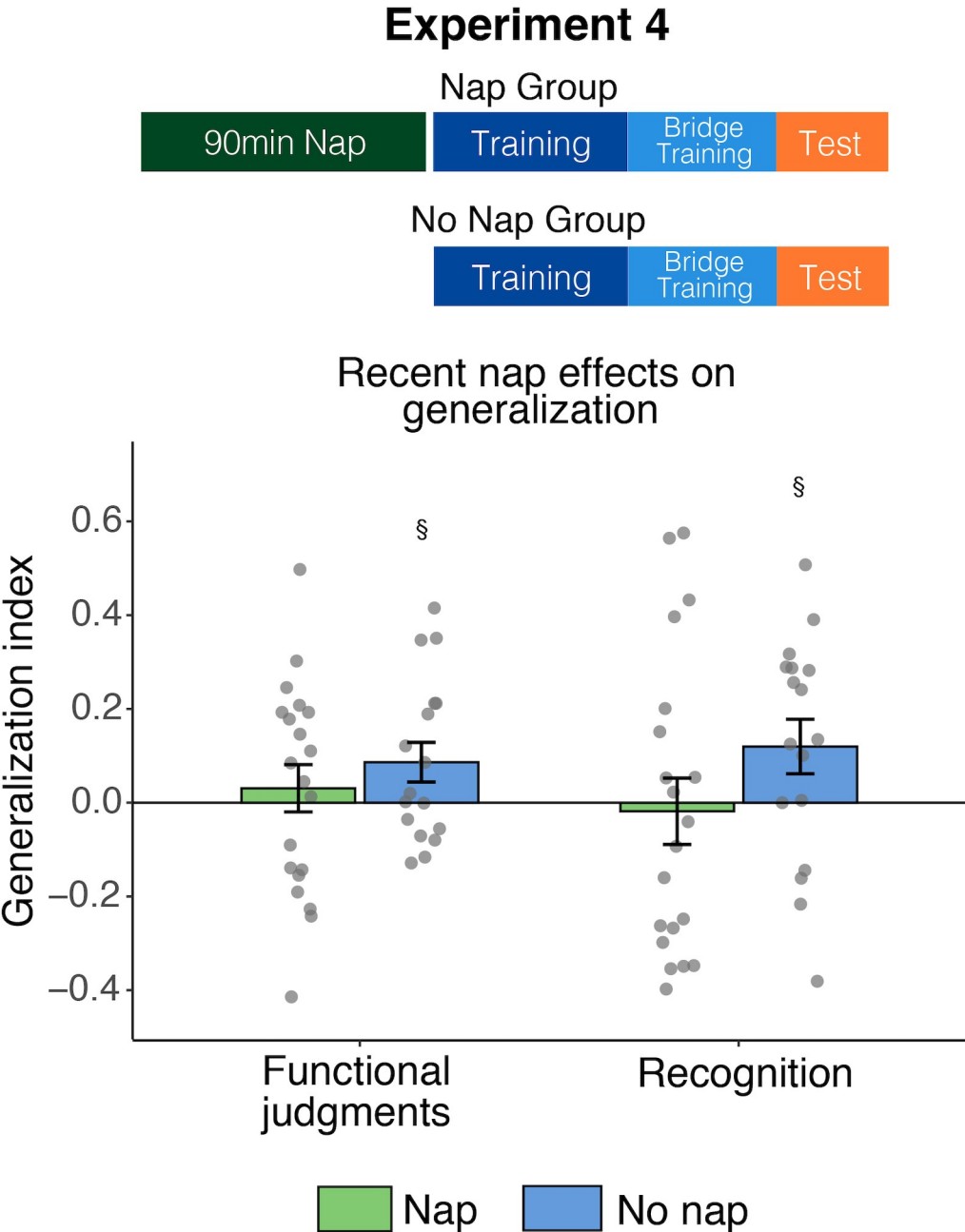

**Fig 3. Effects of having napped recently on generalization.** Generalization is depicted for a group that took an afternoon nap prior to starting the task and a group that did not nap. §$p < .1$.

times of day, such as insight problem-solving on the Remote Associates Test (RAT; [63, 64] and memory for distractors [37, 50]? In this experiment, participants completed these tasks prior to the generalization paradigm. In order to keep the overall experiment length similar with the addition of these two tasks, we also used a simplified generalization paradigm with three instead of five exemplars per category, as well as fewer test trials. We hypothesized that there would be positive across-subject correlations between generalization and these additional tasks.

**Basic measures of performance.** The Evening group provided marginally higher functional ratings than the Morning group for trained satellites ($t[41]$ = -1.93, $p$ = .06; Morning: 0.47±0.05; Evening: 0.63±0.06) and for recognition ($t[41]$ = -1.86, $p$ = .07; Morning: 0.45±0.04; Evening: 0.56±0.05). The Evening group also rated satellites overall as marginally more functional than the Morning group ($t[41]$ = -1.83, $p$ = .07; Morning: 0.19±0.04; Evening:0.32±0.06; recognition: $t[41]$ = -0.02, $p$ = .99). There was no difference between groups in code name memory ($t[41]$ = -0.36, $p$ = 0.72). Overall, basic performance measures did not substantially differ between groups.

**Time of day effects on generalization.** Counter to our previous experiments, we found no effect of time of day on functional judgment generalization ($t[41]$ = -1.79, $p$ = .08) and neither group differed from zero on this measure (Morning: $t[24]$ = -1.03, $p$ = .31, Evening: $t[17]$ = 1.34, $p$ = .20; Fig 4A). Although not significant, the time of day effect was trending in the opposite direction as earlier experiments, with Evening participants exhibiting marginally better generalization than Morning participants. There was no difference in generalization between groups for recognition ($t[41]$ = -1.38, $p$ = .18). While the Evening group was not significantly different from zero on generalization for recognition ($t[17]$ = 0.14, $p$ = .89), the Morning group demonstrated negative generalization on this measure ($t[24]$ = -2.14, $p$ = .04). Regression models with time of day and MEQ interacting as predictors of generalization yielded no significant terms ($p$'s > .11), although there was a main effect of MEQ on recognition generalization ($B$ = 0.02, $t$ = 2.04, $p$ = .05), whereby greater morning preference was associated with better generalization.

**Time of day effects on RAT and distractor priming.** There was no difference between Morning and Evening groups on the RAT ($t[41]$ = 0.63, $p$ = .53) or on the distractor priming test ($t[41]$ = -1.22, $p$ = .23). A regression where we let time of day and MEQ score interact as predictors for each of these tasks yielded no significant terms ($p$'s > .22).

To examine how the tasks relate, we ran correlations between performance on the RAT and distractor priming with measures of generalization separately for the Morning and Evening

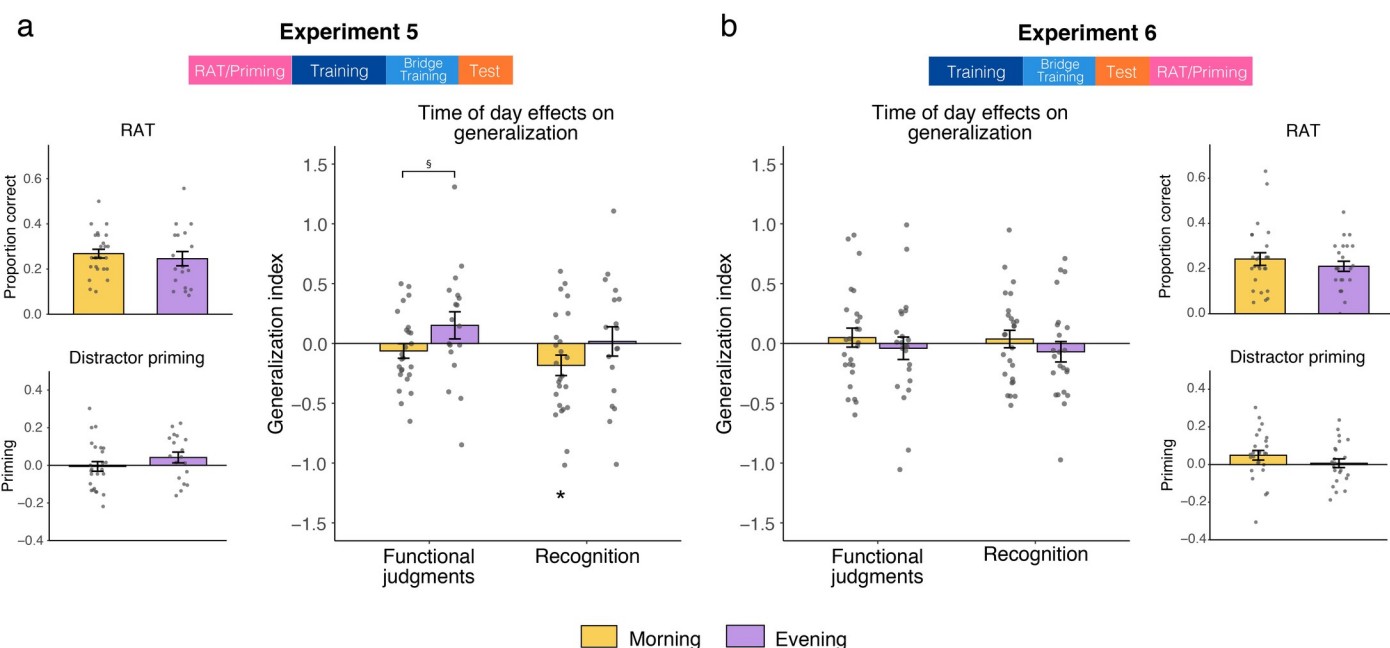

**Fig 4. Time of day effects on RAT, distractor priming, and generalization with a simplified category structure.** Results for three tasks in (a) Experiment 5 and (b) Experiment 6, with different task orderings as indicated in the top schematics. $^*p < .05$, $^\S p < .1$.

groups. None of these correlations were significant (*p's* > .10). There was also no correlation between performance on the RAT and distractor priming task for either group (*p's* > .15).

**Discussion.** In the current experiment, we found no evidence of the morning-time benefit for generalization observed in earlier experiments. The major change in this experiment was the use of a simplified generalization paradigm which had fewer exemplars per category. Given that exemplar variability is important for generalization [81–84], it is possible that this change weakened the strength of the learned category representations, thereby weakening generalization. Another possibility is that administering additional tasks interfered with how participants were performing the generalization task. This is possible given that previous work has shown certain tasks can push participants into more inhibitory states that linger into subsequent task performance [85]. We did not find any effect of time of day, or interaction with chronotype, on the RAT or on the distractor priming, perhaps due to our sample consisting mostly of neutral types (S2 Table). There were no correlations amongst any of these measures, suggesting that these tasks may be supported by different underlying mechanisms.

## Experiment 6

In Experiment 5, we did not replicate the time of day effect on generalization observed in previous experiments. This may have been due to the simplified category structure or to administering other tasks prior to the generalization task. To adjudicate between these possibilities, we ran the same paradigm used in Experiment 5, but administered the RAT and distractor priming test at the end of the study, after the generalization task.

**Basic measures of performance.** There was no difference between groups on the mean slider ratings provided for trained satellites (functional: $t[47] = 0.26$, $p = 0.80$; recognition: $t[47] = 0.12$, $p = 0.91$) or for satellites overall (functional: $t[47] = 0.32$, $p = 0.75$; recognition: t $[47] = 0.48$, $p = 0.64$). Code name memory also did not differ between groups ($t[47] = 0.81$, $p = .42$).

**Time of day effects on generalization.** As in Experiment 5, we found no significant difference between Morning and Evening groups on generalization (functional: $t[47] = 0.73$, $p = 0.47$; recognition: $t[47] = 0.96$, $p = 0.34$; Fig 4B). Neither group differed from zero on either generalization measure (*p's* > .42). Regression models with time of day and MEQ interacting as predictors of generalization yielded no significant terms (*p's* > .30).

**Time of day effects on additional measures.** There was no difference between Morning and Evening groups on the RAT ($t[47] = 0.88$, $p = .38$) or distractor priming ($t[47] = 1.21$, $p = 0.23$). Regression models where we let time of day and MEQ scores interact as predictors for performance on these tasks provided no significant terms (*p's* > .27). No correlations between generalization and RAT or distractor priming were significant for the Morning or Evening group (*p's* > .32). RAT and distractor priming were also not correlated with each other for either group (*p's* > .14).

**Comparing the simplified and original paradigm.** The failure to replicate the clear time of day effect observed in earlier experiments suggests something may be different about learning in the simplified generalization paradigm. To investigate this, we examined how generalization and other performance measures differed as a function of paradigm type. For the following exploratory analyses, we combine Experiment 1 (E1) and E3 (original paradigm), and E5 and E6 (simple paradigm).

First, we assessed how the effect of time of day on generalization differs as a function of paradigm type. A two-way ANOVA revealed a significant interaction between paradigm type and time of day on functional judgment generalization ($F[1,145] = 9.34$, $p = .003$), but no main effect of paradigm type ($F[1,145] = 0.50$, $p = .48$) or time of day ($F[1,145] = 2.45$, $p = .12$).

Recognition generalization yielded similar results, with an interaction between paradigm type and time of day on generalization ($F[1,145] = 4.25$, $p = .04$), and no main effects (paradigm type: $F[1,145] = 0.40$, $p = .53$, time of day: $F[1,145] = 1.01$, $p = .32$). The significant interactions between paradigm type and time of day suggest that the morning-time benefit for generalization may only emerge in the context of strong category structure.

We next examined how other measures of performance varied as a function of paradigm type while ignoring time of day. As expected, given fewer exemplars to remember, the simple paradigm resulted in better code name memory than the original paradigm ($t[147] = -2.90$, $p = .004$; original: 0.50±0.02; simple: 0.59±0.02). Although the mean recognition slider judgments for trained satellites did not differ ($t[147] = 0.18$, $p = .85$), participants in the simple paradigm provided substantially higher recognition judgments for all satellites overall ($t[147] = -4.42$, $p = .00002$; original: -0.05±0.03; simple: 0.09±0.02). These findings suggest that the groups might have differed in their judgments for *novel* satellites. Indeed, we found that participants in the simplified paradigm provided higher recognition ratings for novel satellites ($t[147] = -2.40$, $p = .02$; original: -0.25±0.04; simple: -0.14±.03). This finding is likely to reflect the lack of atypical features in the novel test items for the simplified paradigm. Performance measures that did not vary as a function of paradigm include mean functional judgments for trained satellites and all satellites overall (*p's* > .09).

**Discussion.**   As in Experiment 5, we did not observe a time of day effect on generalization in the simplified version of the paradigm. Thus, the failure to replicate earlier experiments appears related to the change in category structure, as opposed to the change in the order of tasks. Comparisons between the simplified and original paradigm revealed other basic differences in the way participants engaged with the task, with the simplified paradigm resulting in better memory for individual exemplars and a higher tendency to indicate recognition of novel satellites. We again found no time of day effect on the RAT or distractor priming task and no correlations between any of the tasks.

## Experiment 7

We did not observe a time of day effect in the context of simplified category structure, making it difficult to interpret the lack of relationships with the RAT and priming tasks. For the final experiment, we thus returned to the original generalization paradigm and administered the RAT and distractor priming task at the end of the experiment.

**Basic measures of performance.**   Morning and Evening groups did not differ in ratings for trained satellites (functional: $t[77] = 0.10$, $p = 0.92$; recognition: $t[77] = 0.31$, $p = 0.76$) or for all satellites overall (functional: $t[77] = 0.72$, $p = 0.47$; recognition: ($t[77] = 0.34$, $p = .73$). There was no difference between groups in code name memory ($t[77] = 1.46$, $p = .15$).

**Time of day effects on generalization.**   There was no difference between Morning and Evening groups on either measure of generalization (functional: $t[77] = 0.59$, $p = .56$; recognition: $t[77] = -0.93$, $p = .93$; Fig 5). Neither group differed from zero on functional judgment generalization (Morning: $t[40] = 0.19$, $p = .85$; Evening: $t[37] = -0.63$, $p = .53$) or recognition generalization (Morning: $t[40] = -0.15$, $p = .88$; Evening: $t[37] = -0.01$, $p = .99$). MEQ did not interact with time of day in predicting either measure of generalization (*p's* > .40).

**Time of day effects on additional measures.**   Morning and Evening groups did not differ on the RAT ($t[76] = -0.30$, $p = .76$) or distractor priming task ($t[76] = -0.70$, $p = 0.49$; Fig 5). Regression models where we let time of day and MEQ scores interact yielded no significant terms (*p's* > .31). Neither group showed a relationship between RAT and functional judgment generalization (*p's* > .11). The recognition memory measure of generalization exhibited a positive relationship with the RAT for the Morning group ($r = 0.36$, $p = .02$) but not the Evening

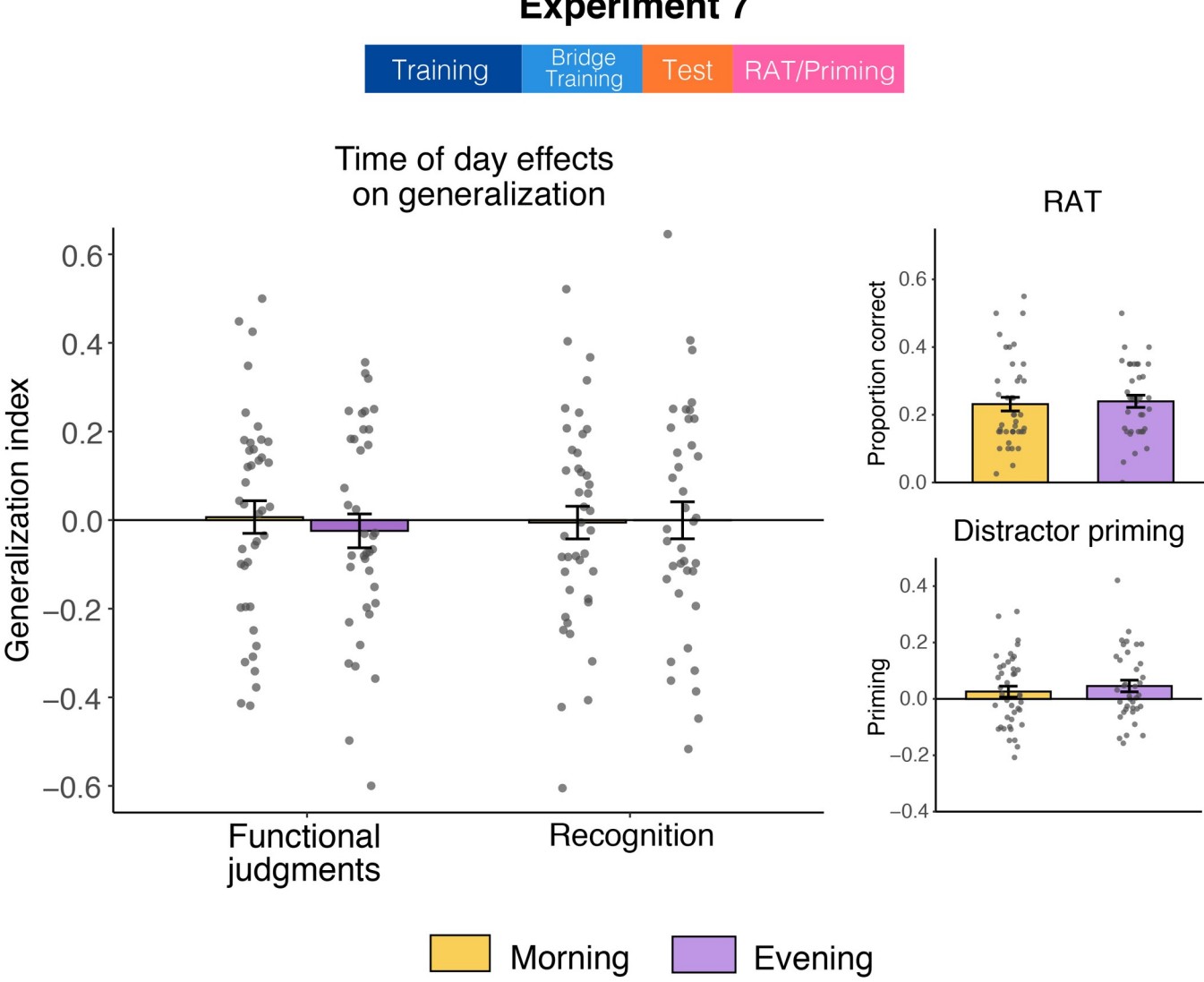

**Fig 5. Time of day effects in Experiment 7.** The generalization paradigm in this experiment was the same as Session 1 of Experiment 1 and Experiment 3. Additional measures, RAT and distractor priming, were administered after the generalization task, with performance shown on the right.

group ($r$ = -0.03, $p$ = .87). This suggests that better performance on the RAT correlated with more generalization-related false memories, but only for the Morning group. Priming did not correlate with either measure of generalization for the Morning or Evening group ($p$'s > .27). RAT and priming were also not correlated with each other for the Morning group ($r$ = -0.08, $p$ = .61), and there was a trending positive association for the Evening group ($r$ = 0.28, $p$ = .09).

   **Discussion.**   Despite returning to the original category structure, we did not find any evidence for a time of day effect on generalization, nor any evidence of generalization at all, in this experiment. In the following results section, we run a series of exploratory analyses to examine potential explanations.

   We again did not find a time of day effect on the RAT or distractor priming task. In this experiment, we found that better insight problem-solving on the RAT was associated with more generalization-related false memories, but only for the Morning group. It is unclear how reliable this relationship is, given that it did not emerge in the previous experiments or in the

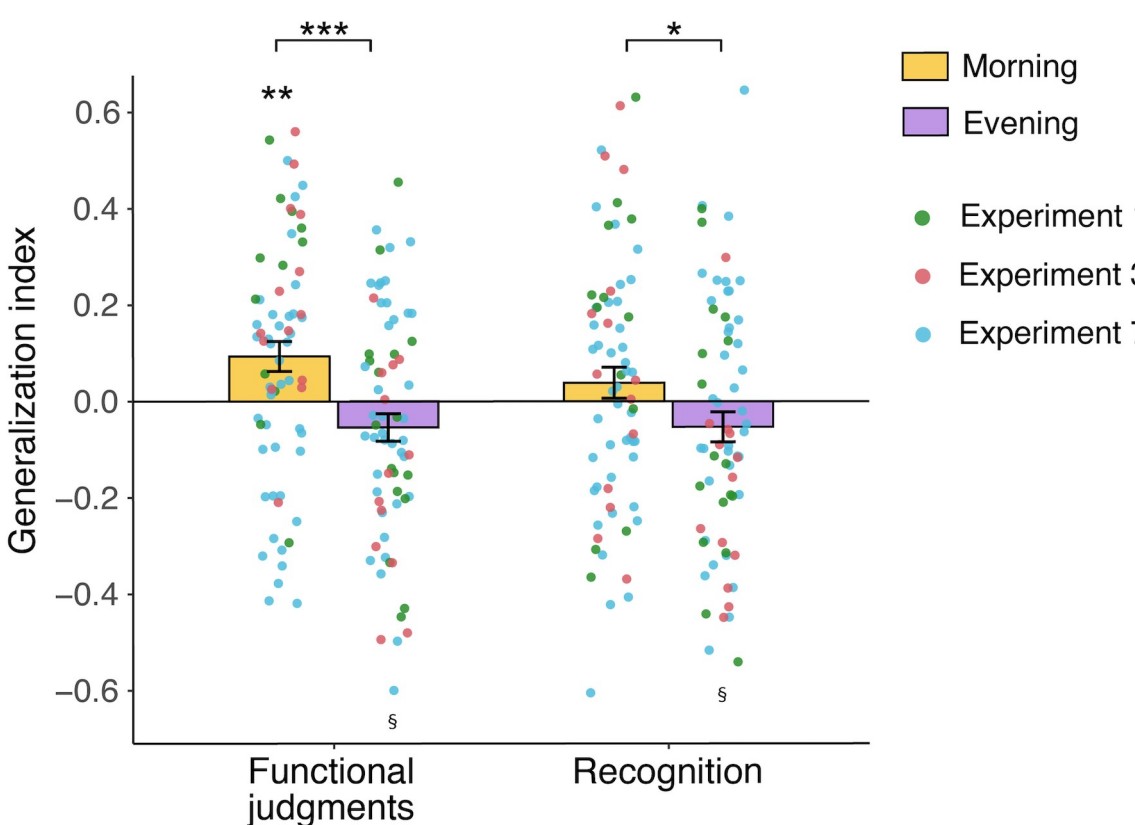

**Fig 6. Overall time of day effects on generalization in experiments with the original paradigm (Experiment 1 Test 1, Experiment 3 Test 1, Experiment 7).** Colored dots reflect individual participants from each experiment. $^{***}p < .001$, $^{**}p < .01$, $^{*}p < .05$, $\S p < .1$.

Evening group in this experiment. Nonetheless, it is worth noting that this finding is consistent with previous work showing that the RAT and false memory are positively associated [86].

**Across experiment analysis: Experiments 1, 3 and 7.** We employed the same generalization paradigm in Experiment 1 (E1), Experiment 3 (E3), and Experiment 7 (E7), yet we did not observe a time of day effect in E7. In this section, we pool data from these experiments to examine the time of day effect on generalization in a large sample size (n = 136; 68 Morning, 68 Evening) and to examine performance differences across experiments that may account for the presence or absence of a time of day effect.

**Overall time of day effects on generalization.** We pooled the data from E7 and Session 1 from E1 and E3. Analyzed together (n = 136), there is a clear effect of time of day on generalization, with better generalization in the morning ($t[134] = 3.49$, $p = .0007$; Fig 6). While the Morning group exhibited reliably positive generalization for functional judgments ($t[67] = 3.01$, $p = .004$), the Evening group demonstrated marginally negative generalization ($t[67] = -1.88$, $p = .06$). There was also a time of day effect on generalization for recognition, with more generalization-related false memories in the morning than evening ($t[134] = 2.04$, $p = .04$). Neither the Morning ($t[67] = 1.20$, $p = .23$) nor Evening group ($t[67] = -1.70$, $p = .09$) on their own were significantly different from zero for this measure, though the Evening group was trending towards negative generalization.

**Differences in time of day effects across experiments.** Despite identical generalization paradigms, E1 and E3 exhibited clear time of day effects whereas E7 did not. To assess how generalization in E7 differed from earlier experiments, we ran a two-way ANOVA on generalization with time of day (Morning, Evening) and experiment (E1, E3, E7) as factors. This revealed a main effect of time of day ($F[1,130] = 13.34$, $p = .004$), with superior generalization in the morning compared to the evening. There was no main effect of experiment on generalization ($F[2,130] = 2.01$, $p = .14$), but there was a significant interaction between time of day and experiment ($F[2,130] = 6.37$, $p = .002$). Post-hoc t-tests revealed that the E7 Morning group's generalization was reliably worse than the Morning groups in E1 ($t[52] = 3.16$, $p = .003$) and E3 ($t[53] = 2.74$, $p = .008$). E1 and E3 Morning groups did not differ from each other ($t[25] = 0.54$, $p = .59$). For the Evening groups, none of the experiments differed from any of the others ($p's > .12$). Overall, this analysis suggests that it is specifically the Morning group that performed worse at generalization in E7 relative to the earlier experiments. The same two-way ANOVA above, but with the recognition measure of generalization, yielded similar effects (time of day: $F[1,130] = 4.31$, $p = .04$; experiment: $F[2,130] = 0.52$, $p = .60$; interaction: $F[2,130] = 3.61$, $p = .03$). Post-hoc comparisons between groups did not yield any differences ($p's > .10$), but the Evening group exhibited better generalization in E7 relative to E3 ($t[49] = -2.32$, $p = .02$).

**Differences in other performance measures across experiments.** We next assessed differences in other measures of performance that might provide insight into why the time of day effect was not observed in E7. Generalization in E1 and E3 did not differ, so we collapsed them together (referred to now as E1&3) to compare them to E7. Designed as sleep studies, these earlier experiments had fewer participants than E7, so combining them also helps to better match sample size.

In the following two-way ANOVAs, we assess the effect of time of day and experiment, as well as the interaction between them, on various measures of performance. We identified several measures that differed across experiments that suggest differential task engagement. Participants in E7 ($F[1,132] = 4.36$, $p = .04$), as well as evening participants ($F[1,132] = 4.88$, $p = .03$) tended to rate satellites overall as less functional (no interaction: $F[1,132] = 1.81$, $p = .18$). Furthermore, participants in E7 ($F[1,132] = 4.27$, $p = .04$) rated the subset of *trained* satellites as less functional, which were the satellites that participants were explicitly told were functional. There was no main effect of time of day on the ability to rate trained satellites as functional ($F[1,132] = 2.01$, $p = .16$). Although the interaction did not reach significance ($F[1,132] = 2.56$, $p = .11$), we ran post-hoc t-tests to better understand the data. This revealed that the Morning group in E7 rated trained satellites as significantly less functional than the Morning group in E1&3 ($t[66] = 2.44$, $p = .02$; E1&3: 0.68±0.04; E7: 0.54±0.04), but Evening groups across experiments did not differ from each other ($t[66] = 0.37$, p = .71) (S3 Fig).

We also examined possible differences in slider strategy use by assessing variability in the overall slider responses (calculated as standard deviation) for each participant. For recognition judgments, participants in E7 exhibited reduced slider response variability ($F[1,132] = 9.41$, $p = .003$). A significant interaction between experiment and time of day ($F[1,132] = 6.77$, $p = .01$) revealed that the Morning group in E7 ($t[66] = 4.01$, $p = .0002$), but not the Evening group ($t[66] = 0.36$, $p = .72$), provided these less variable responses. For functional judgments, similar trends were observed but did not reach significance ($p's > .13$). There were no main effects of time of day on either of these measures ($p's > .13$). As participants were instructed to make use of the full range of the response slider, these analyses suggest participants in E7 were less engaged with the task.

Reduced engagement with the task in E7 might also manifest in response times. Indeed, median response times on the slider judgments were significantly faster in E7 than E1&3

(functional slider: $F[1,132] = 14.43$, $p = .0002$; recognition slider: $F[1,132] = 9.59$, $p = .002$). Time of day and interactions in these models were not significant ($p's > .23$). Basic t-tests comparing these response time measures between E7 and E1&3 further established that E7 participants were responding much faster on both slider judgments (functional slider: $t[134] = 3.76$, $p = .0002$; recognition slider: $t[134] = 3.04$, $p = .003$; S4 Fig).

**Accounting for trained functional judgments in time of day effects on generalization in E7.** On E7 data alone, we ran regressions where we let trained functional judgments and time of day interact in predicting generalization. None of the terms reached significance for the functional judgment generalization model ($p's > .19$), although there was a marginal interaction between time of day and functional judgments ($B = -0.37$, $t = -1.71$, $p = .09$). For recognition generalization, there was a significant interaction between trained functional judgments and time of day in predicting generalization ($B = -0.32$, $t = -2.93$, $p = .005$; S3 Fig). Notably, accounting for trained functional judgments, a time of day effect on generalization does emerge ($B = 0.1$, $t = 2.71$, $p = .008$), with better generalization in the morning. Consistent with the interaction, there was no main effect of trained functional judgments on generalization ($B = 0.07$, $t = 0.66$, $p = .51$). Post-hoc correlations revealed that trained functional judgments positively correlated with generalization for the Morning group ($r = 0.44$, $p = .004$), but there was no significant relationship for the Evening group ($r = -0.22$, $p = .18$). Though these analyses are exploratory, they suggest that the time of day effect on generalization (at least for the recognition memory measure) might emerge only when morning participants can make stronger functional judgments of the trained satellites. This may explain the lack of time of day effect in E7, as Morning participants in E7 provided weaker functional judgments for trained satellites relative to Morning participants in E1&3. While there were other measures that differed in E7 from E1&3 described above, none of these yielded significant terms in these models.

**Discussion.** Collapsing across experiments with matched paradigms (n = 136), there is a clear morning-time benefit for generalization. A comparison across experiments revealed that participants in Experiment 7, relative to Experiments 1 and 3, appear less engaged with the task across multiple measures, including less variable slider responses, taking less time to consider responses, and a reduced ability to rate trained satellites as functional (which were explicitly defined to be functional). In Experiment 7, time of day effects on generalization (at least for recognition) only emerged in Morning participants who were able to effectively rate trained satellites as functional. Though exploratory, these analyses suggest that the lack of replication of the time of day effects in Experiment 7 may be due to this cohort of participants taking the task less seriously; they had faster and less variable responses, in addition to weaker functional judgments for trained satellites (which were defined for the participants to be functional, suggesting poor instruction-following).

## General discussion

Across seven experiments we examined the influence of time of day and sleep on generalization in a category learning paradigm that depends on the integration of previously separate memories. After learning about three distinct categories of novel "functional" satellites, participants briefly encountered bridge satellites that combined features from two of the initially learned categories. We then assessed how participants generalize judgments of functionality and recognition to never-before-seen bridge exemplars that similarly combine features from these two categories.

Counter to our expectations, we found no evidence that sleep facilitates generalization in this paradigm (Experiments 1–3). In lieu of a sleep effect, we found a time of day effect, with

better generalization in the morning than the evening (Experiments 1 and 3). Indeed, even when Evening participants were provided with extensively more bridge training than Morning participants, the Evening group performed no better than the Morning group (Experiment 2). This time of day effect did not appear to be due to the Morning group having slept more recently, as napping provided no benefit to subsequent generalization (Experiment 4). Taken together, these experiments provide evidence that time of day impacts this form of generalization. In subsequent experiments that implemented a simpler category structure, we did not observe the time of day effect (Experiments 5 and 6). We also observed no effect in a final experiment using the more complex paradigm, but found some evidence to suggest that this participant sample engaged less with the task (Experiment 7). The failure to replicate time of day effects in these experiments brings into question the replicability and generalizability of these time of day effects. Nonetheless, collapsing across the three studies with matched paradigms (n = 136) there is a strong morning-time benefit for generalization, which we believe is worthy of further investigation.

## Evidence that generalization is better in the morning

Several cognitive processes are known to fluctuate with the circadian day-night cycle [23, 24]. Our study contributes to this literature by suggesting that generalization across concepts in a novel semantic domain may also be better in the morning than evening. Specifically, we found that participants tested in the morning were not only better able to generalize a property (whether a satellite was "functional") to novel exemplars, but they also exhibited more generalization-related false memories, consistent with findings that generalization occurs alongside false memory formation [68, 69, cf. 70, 71]. Our findings thus illustrate the costs and benefits of generalization: the brain state in the morning that affords us the ability to make useful inferences can also result in false memory for new experiences consistent with one's prior knowledge. We speculate that states of lower inhibition at non-optimal times of day (the morning for most young adults; [32, 34, 39, 59]) might highlight relatively weak connections between memories, facilitating both generalization and false memory. This is consistent with previous work demonstrating that at non-optimal times of day there is both an increased ability to find remote connections between words [63, 64] and increased memory interference [35].

In addition, we found some evidence that the Evening group exhibited a sort of *anti*-generalization: They were more likely to falsely recognize and to endorse as functional novel bridge-*inconsistent* exemplars. Under non-monotonic plasticity accounts [77, 78], this "repulsion" between concepts can occur when related memories are only moderately activated, as might occur in the evening in a state of higher inhibition. Notably, Retrieval-Induced Forgetting, a memory phenomenon that taps into these non-monotonic plasticity and repulsion processes [87, 88], is stronger at optimal times of day and this has been attributed to higher inhibition [48].

Although time of day effects on the retrieval of well-established semantic knowledge are mixed [34, 39, 54–59], two studies have found that individuals rely more on existing semantic knowledge to guide decision-making at non-optimal times of day [89, 90]. Our findings indicate that newly acquired semantic knowledge is also influenced by time of day, in a direction consistent with these studies (as the morning tends to be a non-optimal time for young adults [39, 45, 59]).

Outside of the circadian literature, unexpected morning-time benefits have been observed in sleep studies on tasks that may invoke similar generalization processes, such as category learning [19, 20], generalization of fear extinction memories [15, 21], and perhaps linguistic generalization [22]. There is thus a convergence of evidence of a morning-time benefit for generalization.

## Circadian preference did not modulate time of day effects on generalization

Although the morning is a non-optimal time for young adults at a group level, we did not find that the time of day effect was modulated by individual differences in circadian preference. A major caveat to this finding is that our sample consisted predominantly of neutral chronotypes (S2 Table), whereas studies reporting chronotype-related effects typically sample more densely from extreme morning and evening types [34, 46, 63, 64]. Though neutral type young adults often fail to show time of day effects [59], morning testing has been classified as a non-optimal time for both evening and neutral type young adults [35, 91], so it is likely the morning constituted a non-optimal time for the vast majority of our sample (we had very few morning types). To examine whether an evening benefit could emerge in this task, future work with strong morning types, which could be achieved by sampling older adults [39, 46], would be informative. Another limitation of our ability to speak to this question is that we did not collect chronotypes in Experiment 1 (as we did not originally foresee a time of day effect).

## Time of day effects may be sensitive to category structure and task engagement

We did not find a time of day effect on generalization in a simplified version of our task that involved fewer exemplars per category. Consistent with participants having superior exemplar memory in this version, it is possible that with fewer exemplars participants were simply memorizing the features of each satellite with less sensitivity to the shared structure of the categories, thereby weakening generalization. In support of this possibility, several category learning studies have demonstrated that exemplar variability is particularly important for generalization [cf. 70, 81–84]. Indeed, neither Morning nor Evening groups exhibited reliable generalization in the experiments with fewer exemplars. In sum, the finding suggests that the strength of time of day effects on generalization may depend on features of the task design and task complexity. Another possibility is that the simplified paradigm lacked adequate sensitivity to detect generalization during the test phase, which had less variable items and substantially fewer total trials.

We also did not observe a time of day effect in an experiment where participants appeared to be less engaged with the task, as evidenced by less variable slider responses and faster response times overall. Participants in this experiment also tended to rate trained satellites as less functional. As participants were explicitly told that trained satellites were by definition functional, this behavior suggests a lack of task engagement. Taking this variable into account in exploratory analyses partially recovers the morning-time benefit for generalization. It is possible that participants in this experiment were less invested or motivated given that they were paid less overall than participants in earlier experiments, which involved longer visits or visiting the lab on two occasions.

Several prior studies have demonstrated that factors of a task, such as whether the task is implicit or explicit, can modulate time of day effects [44, 46, 47, 49,50, 53]. The mixed findings we report may reflect these kinds of influences, given that our paradigm is a complex category learning task that may lend itself to the use of multiple strategies.

## Sleep did not promote this form of generalization

Counter to work demonstrating that sleep facilitates generalization [9–11, 13–15, 92], we found no difference in the change in generalization ability across a night of sleep compared to a day awake when initial performance was matched (Experiment 2) or when taken into

account via regression (Experiment 1). Our findings thus contribute to the handful of studies demonstrating instances in which sleep does not facilitate the ability to generalize [3, 4, 12, 16].

Why does sleep fail to promote generalization in our paradigm? By providing minimal opportunity to learn the tested structure during training, we intended for our paradigm to require offline consolidation to generalize. However, the immediate above-chance generalization in our experiments suggests that offline processing is not always necessary for this form of generalization that requires the integration of previously separate memories. In support of this idea, an fMRI study found evidence of integrated neural representations in a period shortly after encountering "bridge" events that linked previously unrelated experiences [18]. Behavioral signatures of a similar form of integration have been linked to neural reinstatement of relevant memories during the bridge learning experiences themselves [17]. It thus appears that inferences dependent upon the integration of previously separate memories can stem from rapid changes in neural representations that occur largely online (during learning or immediately after). However, even in the groups that failed to show reliable evidence of generalization online, there was still no evidence for a benefit for sleep in our experiments.

## Time of day effects did not emerge for the RAT or distractor priming task

Prior work has demonstrated that other aspects of cognition, such as insight problem-solving on the RAT ([63, 64] and memory for distractions (distractor priming; [37, 50] are facilitated at chronotype-defined non-optimal times of day. In our experiments, we did not observe this interaction with chronotype, nor an effect of time of day, on either of these tasks. How might these findings be reconciled? Unlike the prior studies, our sample consisted mostly of neutral chronotypes, making it less sensitive to chronotype-related effects. Indeed, the RAT studies mentioned above did not observe any time of day effects in neutral chronotypes. In addition, a major procedural difference across studies concerns the time at which the evening tests were administered: we tested participants in the late evening (9pm), whereas much of the previous work test in the late afternoon (4–5:30pm [37, 64]). This testing time difference may thus contribute to the discrepant findings, especially given that circadian effects on cognition can be nonlinear across the day [24].

We also did not observe any reliable relationships between these tasks and generalization. A major limitation of this finding is that these measures were only collected in the set of generalization experiments that did not show time of day effects, making the lack of relationships difficult to interpret. Setting this issue aside, to the extent that these tasks are supported by different brain regions [28, 93, 94], a lack of a relationship does not preclude the possibility that they all rely on inhibition as a key mechanism, as circadian-related fluctuations in inhibition may be non-uniform across the brain [95, 96].

## Possible neural mechanisms of generalization and their relationship to inhibition

Although we have suggested that changes in inhibition may underlie the time of day effect on generalization, we did not assess neural inhibition directly in this study. An important avenue for future research will be to more directly establish how time of day changes in neural excitation and inhibition relate to behavioral outcomes. For instance, our research, combined with other behavioral evidence suggesting reduced inhibition in the morning, runs counter to the synaptic homeostasis hypothesis [97] whereby there is a gradual increase in cortical neural excitation across the day [98, 99] that is only renormalized by sleep (i.e., cortical inhibition is strongest in the morning). We also did not find that time since awakening in the morning, or an afternoon nap, related to generalization ability in our experiments. Instead, reduced

morning time inhibition may be better explained by separate circadian neural changes, such as the evening decrease in cortical excitability [100].

An important factor to consider is how circadian fluctuations in inhibition vary by brain region [95, 96]. Generalization in our category learning paradigm is likely dependent on the hippocampus [4, 67, 93, 101–105]. In rodents, hippocampal excitatory-inhibitory balance is strongly modulated by circadian rhythms [106–108]) and neurotransmitter concentrations that alter this balance have been linked to circadian effects on hippocampal memory processes [109, 110]. Although we can currently only speculate as to how neural changes in inhibition relate to behavioral measures of inhibition and generalization in humans, neuroimaging tools that measure the concentrations of excitatory and inhibitory neurotransmitters in different brain regions [111–113] provide an exciting opportunity to investigate this more directly.

In addition, time of day is just one factor that influences inhibition. Changes in the excitatory-inhibitory balance in the brain vary dramatically across the lifespan [114, 115], across different clinical populations [116–118], across sleep stages [119–122], and as a function of task demands [85, 123]. Theories of how inhibition shapes the organization of memories [124–126] will thus be critical for a broad understanding, across many states and populations, of how memories interact to give rise to our ability to generalize [127]. This venture can also provide important insight into the putative neural mechanisms of generalization [67, 103, 104, 128, 129], as different theories make different predictions about the role of excitation and inhibition during memory formation and retrieval [67, 130].

## Broader implications of time of day effects on measures and everyday cognition

Our study serves as a cautionary tale for sleep designs that do not control for time of day—in particular, it is crucial to be able to verify that pre-delay behavior is matched between a sleep and wake group. Time of day effects may also be important to consider more broadly in experimental design and scheduling. For neuroimaging, significant influences of time of day have been observed in both functional [28, 131–134] and structural [135, 136] MRI scans.

Understanding how cognition fluctuates throughout the day is also critical for a better understanding and optimization of everyday behavior. Circadian rhythms interact with nearly every aspect of our daily lives, not least when we attend school or work [137, 138]. While various aspects of cognition are impaired in the morning for young adults [24, 39], leading this time to be considered "non-optimal," the finding that generalization is improved in the morning in this population demonstrates that these effects may depend crucially on the kind of cognition in question. Characterizing these fluctuations and trade-offs can both allow us to better understand the mechanisms of cognition and to better match our lives to the ever-changing states of our brains.

## Supporting information

**S1 Table. Training performance.**
(PDF)

**S2 Table. Survey measures.**
(PDF)

**S1 Fig. Generalization in Test 2.**
(PDF)

**S2 Fig. Change in generalization across sessions (Test 2 generalization–Test 1 generalization).**
(PDF)

**S3 Fig. Functional judgments for trained satellites.**
(PDF)

**S4 Fig. Overall slider response times.**
(PDF)

**S1 File. Data files.**
(ZIP)

# Acknowledgments

We thank Lynn Hasher and Joan Ngo for sharing materials for the distractor priming task, and Kevin Madore for sharing materials for the Remote Associates Test.

# Author Contributions

**Conceptualization:** Robert Stickgold, Anna C. Schapiro.

**Formal analysis:** Marlie C. Tandoc, Mollie Bayda, Eileen Cho, Roy Cox, Anna C. Schapiro.

**Investigation:** Mollie Bayda, Craig Poskanzer, Eileen Cho, Anna C. Schapiro.

**Visualization:** Marlie C. Tandoc.

**Writing – original draft:** Marlie C. Tandoc.

**Writing – review & editing:** Marlie C. Tandoc, Mollie Bayda, Craig Poskanzer, Eileen Cho, Roy Cox, Robert Stickgold, Anna C. Schapiro.

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
