## [Decision Letter · Decision Letter 0]

14 Jun 2021

PONE-D-21-10206

Influences of time of day on generalization

PLOS ONE

Dear Dr. Tandoc,

Thank you for submitting your manuscript to PLOS ONE. After careful consideration, we feel that it has merit but does not fully meet PLOS ONE’s publication criteria as it currently stands. Therefore, we invite you to submit a revised version of the manuscript that addresses the points raised during the review process. In addition to the comments from the reviewers, I have additional suggestion for you to consider. Given that a significant portion of this research specifically focused on the impact of sleep on generalisation, it would be advantageous if the title was modified to include sleep and not just time-of-day. 

We look forward to receiving your revised manuscript.

Kind regards,

Bradley R. King

Academic Editor

PLOS ONE

Journal Requirements:

2) Please modify the title to ensure that it is meeting PLOS’ guidelines (https://journals.plos.org/plosone/s/submission-guidelines#loc-title). In particular, the title should be "specific, descriptive, concise, and comprehensible to readers outside the field" and in this case it is not informative and specific about your study's scope and methodology.

3) Please include captions for your Supporting Information files at the end of your manuscript, and update any in-text citations to match accordingly. Please see our Supporting Information guidelines for more information: http://journals.plos.org/plosone/s/supporting-information.

Reviewers' comments:

Reviewer's Responses to Questions

**Comments to the Author**

1. Is the manuscript technically sound, and do the data support the conclusions?

Reviewer #1: Yes

Reviewer #2: Partly

2. Has the statistical analysis been performed appropriately and rigorously? 

Reviewer #1: Yes

Reviewer #2: No

3. Have the authors made all data underlying the findings in their manuscript fully available?

Reviewer #1: Yes

Reviewer #2: Yes

4. Is the manuscript presented in an intelligible fashion and written in standard English?

Reviewer #1: Yes

Reviewer #2: Yes

5. Review Comments to the Author

Reviewer #1: This is a very long, complex paper which uses a very complicated task to assess whether or not and to what degree people generalize from exemplars. It provides, across a series of studies, a reasonably clear message about sleep (it doesn't seem to influence generalization) and a somewhat less strong message about time of testing (there is greater generalization in the morning than in the evening). The authors do an excellent job of explaining their complex patterns of results and make clear the connections to existing literature (i.e., the references and the explanations) are excellent. The discussion is particularly strong and may well be influential for subsequent work.

what would have made this paper a better paper? If it were possible to focus/limit analyses to DVs relevant to generalization, it would have been a far easier read, especially given the large number of studies included. On the other hand, this is an extremely honest paper which allows readers to see warts in the research process.

Reviewer #2: This study investigates a very interesting phenomena, (specifically, generalization of newly acquired memories) that sleep would appear to be a likely candidate to facilitate/support/enhance. The inclusion of a nap control experiment provides valuable insight, and is something that is not normally part of experimental designs that include sleep, so this welcome addition allows for valuable comparisons between a night of sleep and a nap. Along the same line, additional behavioural experiments to further prole time of day add rigor to the overall report, despite that some effects were not easily replicated, highlighting the inherent complicated nature of studying this phenomena and the pitfalls of overlooking time-of-day effects.

Major comments:

1. One of the main conclusions is difficult to support based on the approach and outcomes, specifically, statements such as: “…suggesting a role for circadian rhythms apart from sleep”. It cannot be determined if circadian rhythms, per se, or other factors linked to time of day, or time elapsed between test and retest explain the effect of time on generalization. This conclusion is also not supported by / consistent with the analyses on chronotype. Suggest removing this or modifying these statements to be more in line with the study design and results.

2. Another main conclusion: “…a state of lowered inhibition in the morning may facilitate spreading activation between otherwise separate memories…” is difficult to substantiate, as “inhibition” was not directly measured in the current study. Suggest framing this more as a possibility to be investigated in the future, rather than an interpretation that can be made from the current study.

3. Null results are difficult to interpret; for which there are many across the series of Experiments. It would be helpful to get a sense of whether the null findings were due to insufficient statistical power, re; ~ 15/group is common in the literature, but often questionable whether sufficient to detect variable/subtle behavioural effects. If power analyses were conducted, they should be presented, otherwise, it would be helpful to know if the study was adequately powered. The pooled analysis presented in Experiment #7 partly speaks to this issue, but not expressly for each individual experiment.

4. In Experiment #1, several correlations between outcome variables were performed, however, it is not clear how many such analyses were performed, and if this was done in any systematic/parsimonious manner and seem overly exploratory the way they are presented. Multiple comparisons may be an issue if many analyses were conducted without any correction. This also applies to subsequent experiments.

5. There are a large number of experiments presented, with similar, but not exactly the same analysis strategies employed, with a very large number of null results presented alongside significant findings. It is easy for the main message to get buried in all this. Summary tables of results might help with the presentation.

6. The conclusion (albeit stated as exploratory), that the “lack of replication of the time of day effects in Experiment 7 may be due to this cohort of participants taking the task less seriously” is unfounded. How was the “seriousness” of the participants ascertained?

Minor comments:

1. While the initial aim of the first 3 experiments was to investigate sleep, and the hypotheses are presented only very briefly in the introduction. The subsequent studies were designed to investigate why no effect of sleep was observed. The Introduction could benefit from presenting specific hypotheses for the subsequent studies.

2. How many participants reached the 1-hour cut-off for the training phase? Performance in the case where they were cut-off due to the time limit may be interpreted differently than those who completed the training in the allotted time.

3. Partial credit on the RAT was given using Levenshtein distance, which is innovative, but perhaps not appropriate, given that the goal of the task is not to assess spelling/typographical errors. i.e., this measure confounds semantic accuracy with typographical precision (which is not of interest in this case). It might be more appropriate to instead manually assess whether errors were typographical, in which case, effectively the correct word (full points), or an incorrect word (no points). If a large number of responses were partial scores, and if in most cases the Levenshtein distance is neither close to a full point nor zero, then this may be problematic. By contrast, this may not be an issue for the priming score in the distractor priming task, re; lexical accuracy is relevant in that case.

4. The results of the earlier experiments were not replicated, which was attributed to the changes in the task design. This is intriguing as it would suggest some sort of dose-response relationship for the shorter version. Suggest mentioning this possibility in the Discussion.

5. The Discussion is very long, particularly given that there are already “interim” discussions for each experiment, and also in light of the large volume of null (and somewhat inconclusive) results. Suggest significantly shortening this section to summarize the main message.

6. PLOS authors have the option to publish the peer review history of their article (what does this mean?). If published, this will include your full peer review and any attached files.

Reviewer #1: No

Reviewer #2: No

---

## [Author Response · Author response to Decision Letter 0]

8 Jul 2021

Please find the response to reviewers in the uploaded document "Response To Reviewers.pdf"

---

## [Decision Letter · Decision Letter 1]

16 Jul 2021

Examining the effects of time of day and sleep on generalization

PONE-D-21-10206R1

Dear Dr. Tandoc,

We’re pleased to inform you that your manuscript has been judged scientifically suitable for publication and will be formally accepted for publication once it meets all outstanding technical requirements.

Kind regards,

Bradley R. King

Academic Editor

PLOS ONE

Additional Editor Comments (optional):

Reviewers' comments:

Reviewer's Responses to Questions

**Comments to the Author**

1. If the authors have adequately addressed your comments raised in a previous round of review and you feel that this manuscript is now acceptable for publication, you may indicate that here to bypass the “Comments to the Author” section, enter your conflict of interest statement in the “Confidential to Editor” section, and submit your "Accept" recommendation.

Reviewer #2: All comments have been addressed

2. Is the manuscript technically sound, and do the data support the conclusions?

Reviewer #2: Yes

3. Has the statistical analysis been performed appropriately and rigorously? 

Reviewer #2: Yes

4. Have the authors made all data underlying the findings in their manuscript fully available?

Reviewer #2: (No Response)

5. Is the manuscript presented in an intelligible fashion and written in standard English?

Reviewer #2: Yes

6. Review Comments to the Author

Reviewer #2: The authors have done a commendable job in replying to all of the comments made in the first round of revision.

7. PLOS authors have the option to publish the peer review history of their article (what does this mean?). If published, this will include your full peer review and any attached files.

Reviewer #2: No

---

## [Editor Report · Acceptance letter]

23 Jul 2021

PONE-D-21-10206R1 

Examining the effects of time of day and sleep on generalization 

Dear Dr. Tandoc:

I'm pleased to inform you that your manuscript has been deemed suitable for publication in PLOS ONE. Congratulations! Your manuscript is now with our production department. 

Kind regards, 

on behalf of

Dr. Bradley R. King 

Academic Editor

PLOS ONE